# Atopic Dermatitis: Disease Features, Therapeutic Options, and a Multidisciplinary Approach

**DOI:** 10.3390/life13061419

**Published:** 2023-06-20

**Authors:** Liborija Lugović-Mihić, Jelena Meštrović-Štefekov, Ines Potočnjak, Tea Cindrić, Ivana Ilić, Ivan Lovrić, Lucija Skalicki, Iva Bešlić, Nives Pondeljak

**Affiliations:** 1Department of Dermatovenereology, Sestre Milosrdnice University Hospital Center, 10000 Zagreb, Croatia; jms@globalnet.hr (J.M.-Š.); ivaaabukvic@gmail.com (I.B.); 2School of Dental Medicine, University of Zagreb, 10000 Zagreb, Croatia; 3Institute for Clinical Medical Research and Education, Sestre Milosrdnice University Hospital Center, 10000 Zagreb, Croatia; 4Faculty of Medicine, Josip Juraj Strossmayer University of Osijek, 31000 Osijek, Croatia; ilic.ivana2212@gmail.com (I.I.);; 5Department of Dermatology and Venereology, General Hospital Sisak, 44000 Sisak, Croatia

**Keywords:** atopic dermatitis, atopic eczema, multidisciplinary, pathogenesis, treatment, quality of life, therapy, psychological aspects, JAK inhibitors, dupilumab

## Abstract

The latest findings regarding AD pathogenesis point to an impaired function of the epidermal barrier, changed immune response, colonization of the skin by microorganisms, and certain psychological factors among other causes/triggers. The inflammatory response of AD patients is mainly associated with the activation of T cells (Th2 cells predominate), dendritic cells, macrophages, keratinocytes, mast cells, and eosinophils. Therapy usually involves medical evaluations and adequate management including treatment of concomitant diseases (e.g., allergies and infections), patient education and nursing care, psychological support, and nutritional consultations, which are organized through specific programs and structured educational groups. Systemic AD therapy includes conventional systemic treatment (cyclosporine, methotrexate, azathioprine) and new, specific drugs, interleukin inhibitors (e.g., dupilumab) and JAK inhibitors (baricitinib, abrocitinib, upadacitinib, etc.). Since many AD patients are affected by various psychological factors and comorbidities, they should be assessed and managed through a multidisciplinary approach, involving different professions (psychologists, ear–nose–throat specialists, pulmonologists, allergologists, immunologists, nutritionists, pediatricians, gastroenterologists, psychiatrists (when necessary), and others). A multidisciplinary approach provides better coping strategies and improves control over the disease, patient adherence to therapy, and quality of life. It also has a positive influence on family quality of life while at the same time making more efficient use of dermatology healthcare resources, reducing the economic burden on both patients and society.

## 1. Introduction

Atopic dermatitis (AD) is predominantly characterized by eczema, dry skin, and itching (Figure 1) [1,2,3,4,5]. These symptoms are age-dependent and often associated with other atopic diseases (allergic asthma, allergic rhinitis, and food allergies). Numerous hereditary and environmental factors, and their mutual interactions, participate in the development and clinical manifestations of AD, which can vary significantly in appearance, intensity, and course. The latest findings regarding AD pathogenesis point to a disturbance in the function of the epidermal barrier, a disruption of the immune response, colonization of the skin by microorganisms, an increased tendency toward infection, and certain psychological factors among other causes/triggers.

The prevalence of AD has doubled or tripled over the last 30 years in industrialized countries. It is estimated that 15% to 30% of children, and 2% to 10% of adults, are affected [4,5,6,7]. AD typically occurs in the first year of life (in about 60% of cases) but may appear at any age, and its complete remission has been observed in 70% of children before adolescence. Early onset of the disease, familial associations, and early allergen hypersensitivity are all risk factors for a prolonged disease course and persistence into adulthood [8,9].

## 2. Basic Factors Involved in the Etiopathogenesis of Atopic Dermatitis

The etiopathogenesis of AD is complex and poorly understood, but it is thought to be caused by a complex interaction of genetic and environmental factors [1,4,10,11,12,13,14,15,16]. Thus, the pathogenesis of AD includes epidermal barrier dysfunction, immune dysregulation, and alteration of the skin microbiome, all of which can be altered by genetic and environmental factors. In particular, the key pathomechanism of AD development is the disruption of the epidermal barrier’s function and a changed immune response. Epidermal barrier dysfunction permits easier entry of microbes, allergens, and irritants, which trigger activation of immune response and release of pro-inflammatory cytokines. So, impairment of epidermal barrier function leads to immune dysregulation of innate and adaptive responses to environmental stimuli. It is known that the adaptive response is represented by Th2 cytokines, namely IL-4, IL-5, IL-13, and IL-31; eosinophilic activation; and production of allergen-specific IgE. In addition, disruption of the epidermal barrier’s function is related to the filaggrin protein, which is, along with keratin, the most important structural protein of the stratum corneum. A mutation of the filaggrin gene on chromosome 1q21 is common in AD patients, and it is significantly associated with disruptions to the structure and function of the epidermal barrier. This leads to an increase in transepidermal water loss (TEWL) and decreased synthesis of natural moisturizing factors (NMFs), which are produced in the process of filaggrin deamination [10,11]. In addition, the pH of AD patients’ skin is increased, and this increase in skin pH stimulates the activity of serine protease and kallikrein, which weakens the action of the enzymes responsible for ceramide synthesis. Decreased ceramide production is also caused by increased sphingomyelin deacylase activity—high expression of sphingomyelin deacylase indicates ceramide deficiency—and ultimately improves the possibility of environmental allergens penetrating deep into the damaged epidermal barrier [11,12,13].

In the epidermal skin barrier of AD patients, there is a correlation between immunological and structural disturbances. Genetic factors play a role in the development of immune barrier dysfunction, specifically chromosome 5q31-33 as it contains a group of family genes that code for the synthesis of Th2 cells that secrete the interleukins IL-3, IL-4, IL-5, and IL-13 and granulocyte-macrophage colony-stimulating factor (GM-CSF), which are important in the inflammatory response [14,15,16]. The deficiency of filaggrin increases the expression of keratinocyte-derived thymic stromal lymphopoietin (TSLP), which is crucial for the onset of skin inflammation in AD. So, TSLP activates dendritic cells that stimulate inflammatory Th2 cell differentiation and cytokine production. It seems that in AD patients, the increased Th2 cytokine expression causes an increase in serine proteases. In addition, tumor necrosis factor (TNFα) together with Th2 cytokines (IL-4, IL-13, IL-31) enhances the release of TSLP and limits the synthesis of long-chain free fatty acids, which allows damage to the skin’s lipid barrier to occur [12]. In addition, interferon-γ (IFN-γ) is involved in keratinocytes’ release of cytokines and chemokines, as well as IL-31, which exacerbates itch and is associated with mechanical trauma (e.g., the impulse to scratch the skin) [17,18]. In addition, exposure to environmental allergens and bacterial infections significantly increases the expression of TSLP, IL-25, and IL-33 and enhances the Th2-dependent response that triggers various interrelated events, leading to a chronic and recurrent AD course [19].

The inflammatory response of AD patients is mainly associated with the activation of T cells, dendritic cells, keratinocytes, macrophages, mast cells, and eosinophils. In acute AD lesions, the infiltrate predominantly consists of CD41+ cells, antigen-presenting cells (APCs) (Langerhans cells, other dendritic cells, and macrophages) with IgE bound to surface receptors. Thus, in the dermis, T cells predominate, with an increased number of CD4+ cells [20]. On the other hand, with chronic lesions, there are considerable dermal accumulations of collagen with a decreased T-cell count and a rich cellular infiltrate of eosinophils and macrophages. In inflammatory skin lesions, the T cells stimulate effector cytokines and induce keratinocyte activation and apoptosis. In the peripheral blood vessels of AD patients, we see natural regulatory T cells (Tregs) (CD4+, CD25+, FoxP3+) with regular immunosuppressive activity allowing Tregs to interfere with Th1 and Th2 responses. Mutations of the nuclear factor of Tregs cause immune dysregulation characterized by increased IgE levels, food allergies, and eczema. According to research data on the role of Tregs in AD, after stimulation with staphylococcal superantigen (enterotoxin type B), Tregs lose their immunosuppressive activity, which can exacerbate skin inflammation [13].

A damaged epidermal barrier increases the local expression of pro-inflammatory cytokines and chemokines, which bind to specific vascular endothelial receptors, and this facilitates the migration of pro-inflammatory cells that form AD skin infiltrate [21]. Thus, impaired stratum corneum is associated with filaggrin and other barrier protein deficiencies, elevated levels of endogenous serine proteases, and abnormal lipid composition which leads to increased stratum corneum permeability. All of these lead to impairment of epidermal barrier function and a vicious cycle of immune dysregulation. In inflammatory AD lesions (acute and chronic), a significant increase in IL-4, IL-5, and IL-13 can be observed. Even in apparently healthy skin, an increased expression of IL-4 and IL-13 (Th2 cytokines) is seen, which is not true for other cytokines such as IL-5 or IFN-gamma. Some of the above-mentioned pro-inflammatory cytokines, in combination with costimulatory molecules, initiate the differentiation of B cells into plasma cells that produce IgE antibodies. In addition, in AD lesions, receptors for IgE have been detected on the surface of myeloid dendritic cells, i.e., Langerhans cells and inflammatory dendritic epidermal cells (IDECs). The IDEC cells are responsible for synthesizing and releasing Th1 cytokines, while Langerhans cells are crucial for triggering the allergic immune response and controlling the antigen presentation to T cells, which in turn migrate into the epidermis and release numerous cytokines that induce keratinocyte apoptosis [19]. Stimulation of receptors on IDECs promotes the release of many inflammatory factors, which contribute to the amplification of the immune response. In skin lesions of AD patients, the expression of IL-31 is markedly increased.

Chronic inflammation in AD is associated with the dominance of Th1 cytokines (IL-12, IL-18, IL-11, TGF-β, and IL-31). At the same time, IL-5 is responsible for the maturation of eosinophils and the prolongation of their survival. Aside from skin lesions/eczema, chronic inflammation includes, or can lead to, concomitant itch, caused by various factors such as IL-31 produced by Th2 cells, an increased histamine release, colonization of the skin by *Staphylococcus aureus*, and other secondary issues. Thus, IL-31 is a Th2 cytokine that is highly expressed in AD lesions. Exposure to a staphylococcal antigen leads to increased expression of IL-31, bridging a direct connection between staphylococcal exposure and pruritus. *Staphylococcus aureus* stimulates mast cell degranulation, histamine release, and Th2 inflammation. So, the Th2 response further damages the skin barrier, worsens itching, and facilitates dysbiosis in favor of *Staphylococcus aureus*. In addition, Langerhans cells and inflammatory epidermal dendritic cells with specific IgE attached to the high-affinity IgE receptor, along with dermal dendritic cells, capture allergens and antigens in the affected skin. This triggers an immune response that activates sensory nerves via cytokines such as IL-4, IL-13, and IL-31, resulting in itchiness. Generally, the main cytokines involved in the development of AD lesions are Th2 cytokines, primarily IL-4 and IL-13, and current therapy options are often based on their suppression. It is known that type 2 cytokines, IL-4 and IL-13, have a crucial role in Th2 cell differentiation, IgE production, and eosinophil recruitment. They contribute to itch and are considered relevant to chronic pruritus in patients with AD. The predominant effect of increased IL-4 and IL-13 expression is the recruitment of eosinophils and basophils to tissues, along with heightened activation of Th2 cells and increased production of IgE. Additionally, IL-4 plays a role in the activation of Th2 cells via JAK-STAT signaling, leading to the production of various cytokines and chemokines. Knowledge of these pathogenetic processes is crucial for understanding AD and the creation of more effective therapy options.

## 3. Skin Itching and Related Factors in Atopic Dermatitis

Aside from eczema, the main symptom of AD is severe itching, which greatly affects patients’ daily activities, productivity, and sleep, reducing their overall quality of life. The pathophysiology of itch lies in the interaction between keratinocytes, the immune system, and non-histaminic nerve endings. Thus, Th2 cells, eosinophils, and neutrophils play important roles in activating the itch circuit, and mast cells release pro-inflammatory cytokines and peptides. Itch is mediated by the interaction of non-myelinated C-nerve terminals and myelinated A-delta nerve terminals [4,22]. One of the typical mediators of itch is IL-31 (produced by Th2 cells), and its increased concentration has been detected in AD lesions [5,21]. Thus, IL-31 binds to the IL-31 receptor called IL-31A (IL-31RA), which stimulates the branching and elongation of nerve endings, which then worsens itching [5,17]. In addition, itching leads to scratching, which itself intensifies itching—a process known as the AD “itch–scratch cycle”. In addition, keratinocytes release other itch-related cytokines, such as the alarmin TSLP, which stimulates the Th2 immune pathway and the production of additional itch-inducing cytokines, thus closing the cycle of itching.

Numerous endogenous and exogenous itch-related factors such as leukotrienes, interleukins, and endothelin are important, in addition to those mentioned above [23]. Atopic dermatitis patients are incentivized to itch through simulation, which has not been recorded in healthy adults (this phenomenon is known as peripheral itch sensitization) [22]. In the skin of AD patients with a distinct clinical picture—even with clinically mild non-itchy eczema—a greater density and thickness of nerve endings was found compared to the control group, which may explain the increased itching sensation they experience [23]. Other AD-related factors can also exacerbate itch, such as emotional stress, a stressful lifestyle, a lack of quality sleep, and consumption of alcoholic beverages [4]. Concerning histamine, its role in itching and the above process is unclear. When the histamine receptors H1R and H4R are simultaneously blocked, itch and skin inflammation are calmed more effectively than when these receptors are blocked individually. Some literature data show that non-sedative antihistamines do not significantly suppress itch in AD patients, indicating the so-called non-histaminic pathogenesis of itch in AD [4]. However, the insight into many above-mentioned factors is a base for anti-inflammatory therapy (from topical corticosteroids to complex biologic medicines) which plays a role in alleviating itch as the main AD symptom [6,23].

Atopic dermatitis’ concomitant itch can be attributed to a range of causes, including the colonization of the skin by *Staphylococcus aureus* (i.e., the activity of the superantigen *Staphylococcus aureus*) and changes in the skin’s microbiome (AD is among many diseases associated with alterations in the microbiome, as confirmed by many studies) [7,24]. The skin’s microbiome includes all microorganisms found on the free epidermal surface (primarily bacteria, but also all viruses, protozoa, and fungi); the composition depends on the individual and on the specific area/localization of the skin. Various factors influence the skin’s microbiome: skin thickness, the effect of UV radiation, temperature, moisture, the composition and amount of sebum, pH heterogeneity of the skin, etc. [24]. Microorganisms living on the free surface of the skin coexist with the host and interact with each other. The microbiome is involved in metabolic processes and maintains the immune system by mediating innate and acquired immunity, and its composition (characterized by diversity/heterogeneity) protects against the invasion of pathogenic microorganisms and maintains the stability of the epidermal barrier. Disruptions to, and the dysfunction of, the skin microbiome (dysbiosis) are what allow AD to occur and are seen in every case [7,8,21,24,25]. This means that the skin microbiome composition of AD patients (even those with mild eczema) differs from that of people with healthy skin—AD patients have fewer commensal bacteria such as *Streptococcus, Corynebacterium*, and *Cutibacterium* and an increased number of *Staphylococcus aureus* [25,26]. Changes in the skin microbiome precede clinical AD manifestations, indicating the need to maintain and care for the physiological skin microbiome with the daily use of neutral creams and emollients to avoid AD manifestations. Topical corticosteroids and calcineurin inhibitors can also restore the skin’s epidermal barrier and maintain a normal skin microbiome [15,16]. A decrease in skin microbiome diversity correlates with the severity of AD. The Gram-positive bacterium *Staphylococcus aureus* also plays an important role. as it can colonize inflamed lesions, leading to reduced filaggrin proteins, a higher skin pH, and a reduced production of antimicrobial peptides [10].

## 4. Clinical Manifestations of Atopic Dermatitis and Diagnostic Criteria

The clinical picture of AD is very diverse and depends on patient age, localization, disease severity, duration, and potential complications (e.g., secondary infections). Eczematous AD lesions commonly involve itch and can be acute, subacute, or chronic, and they can range from very mild to very severe clinical forms [26]. The course of AD is mostly chronic and relapsing, with phases of remission and exacerbation of varying length. In some patients, AD occurs seasonally, with common exacerbations in spring and autumn, which may be accompanied by allergic rhinoconjunctivitis or allergic asthma. It is commonly known that AD patients are genetically predisposed to produce allergen-specific IgEs following exposure to those allergens, which at certain ages may manifest as a typical sequence of AD, allergic rhinitis, asthma, and food allergy (the “*atopic march*”). Although allergies are confirmed in the majority of AD patients (extrinsic AD), sometimes they are not, which indicates intrinsic AD [27]. However, it is necessary to take into account AD patients’ comorbidities. For example, AD patients are more prone to developing chronic rhinosinusitis, nasal polyps, oral allergy syndrome, food-induced urticaria/anaphylaxis, eosinophilic esophagitis, ichthyosis vulgaris, and ophthalmic conditions [28,29,30]. Other comorbidities sometimes seen in AD patients, but with controversial study results on their association with AD, are obesity, metabolic syndrome, cardiovascular disease, and anemia, as well as an increased risk for cancer (e.g., lymphoma and skin cancer).

Atopic dermatitis is a chronic disease with acute flares and remissions, and its manifestations can vary [31]. In acute AD, lesions are erythematous, frequently exudative and weeping, sometimes with blisters that rupture, leading to erosions and, finally, crusts. Over time, the skin becomes chronically changed, thickened (lichenified), and sometimes scaly. Some typical signs of AD are keratosis pilaris, nipple eczema, white dermographism, hyperlinear palms, and a Dennie–Morgan fold (a fold below the lower eyelid) [31]. In the early stages (at birth or shortly after), AD manifestations may be similar to infantile seborrheic dermatitis (with lesions on the scalp and folds), and skin is often dry and rough. Usually, skin changes/lesions appear on the face (especially the cheeks) and flexural parts of the limbs. Due to frequent rubbing, lesions may appear on some exposed parts such as the backs of the hands, while lichenification, madarosis, nodular prurigo, and other manifestations can also occur. Lesions are not always confined to localizations and may be extensive. As children grow, the localizations and manifestations typically change—lesions appear on the flexor parts of limbs (the elbows, wrists, ankles, knees, etc.) or may occur on the head (perioral region and chin). Due to itching, scratching, and repeated rubbing, the skin becomes lichenified (thickened and dry). Other possible manifestations, especially in school-age children and adolescents, are *pityriasis alba*, pompholyx, cheilitis, *keratosis pilaris,* dirty neck, facial pallor, ichthyosis, ocular/periocular lesions, and others. During school-age years, AD mostly improves clinically, although disrupted skin barrier function persists. In adulthood, the disease may manifest in various forms—it may last and cause flexural lesions, or lesions can become diffuse. Some AD patients who have suffered from childhood may see their condition completely resolve, although it can always recur (e.g., on the hands *due to daily activities*). Sometimes AD occurs for the first time in adulthood (“adult-onset” AD) [31].

The diagnosis for AD is based on the patient’s clinical picture, as there are no specific biochemical or histological parameters. The universally accepted diagnostic criteria for AD by Hanifin and Rajka (adopted 1980) include 4 essential/major and 23 secondary/minor criteria. To make a diagnosis, at least three basic and three secondary criteria must be met. The sensitivity of these criteria ranges from 87.9% to 96.0%, and the specificity ranges from 77.6% to 93.8% [32,33]. The basic criteria are as follows: typical skin lesions and their distribution, lichenification of the folds and linearity of the skin in adults, involvement of the face and extensor sides in infants and young children, a chronic or chronic recurrent course of dermatitis, a personal or family history of atopic diseases (asthma, allergic rhinitis or AD), and pruritus. To simplify the classification, the British working group revised the diagnostic criteria in 1994, in addition to itching as the primary symptom, three or more other symptoms were added: onset of the disease before the age of two (if the child is not younger than four years); previous involvement of the skin folds (or cheeks in children younger than ten years), lesions in the folds; data on generalized skin dryness; and other atopic diseases in the personal history (or a family history of atopic diseases for children younger than four) [34]. In its 2014 guidelines, the American Academy of Dermatology proposed revisions to the 2003 Hanifin and Rajka diagnostic criteria, which include primary, major, and associated disease criteria [35]. Thus, in addition to the primary diagnostic criteria (pruritus and eczema with typical morphology and age-dependent distribution), there are major criteria (early onset of disease, a personal or family history of atopic disease, and dry skin) and associated criteria (atypical vascular reaction (e.g., facial pallor, skin dryness, white dermographism], *keratosis pilaris, pityriasis alba*, palmar hyperlinearity, ichthyosis, ocular and periocular lesions, lesions in other regions (e.g., perioral and periauricular lesions), lichenification, and excoriations). To establish a diagnosis of AD, the primary criteria must be fulfilled, while major criteria support the diagnosis and are observed in most patients (associated criteria also help establish the diagnosis but are not specific).

To assess AD severity, various measurement tools exist, but they are mainly used in clinical trials and rarely in clinical practice. Of the 16 instruments accepted and tested, the Eczema Area and Severity Index (EASI) and the Scoring Atopic Dermatitis (SCORAD) are rated the highest [36,37]. The EASI has reasonable validity, speed of response, internal consistency, and interobserver and interoperator reliability, but unclear interpretability and feasibility. In contrast, the SCORAD has good validity, speed of response, interoperator reliability, and interpretability and unclear interobserver reliability [38]. To standardize and validate them, the Global Harmonising Outcome Measures for Eczema initiative (HOME) was established, highlighting the disease’s clinical symptoms, patients’ quality of life, and patients’ long-term follow-up as crucial quality parameters for measuring the severity of AD [39]

## 5. Available Treatment Possibilities for Atopic Dermatitis

The field of AD treatment in dermatology is currently experiencing a renaissance, as regulatory institutions worldwide have already approved new medications for AD patients, and many other drugs are still in the clinical trial phase. The treatment of AD includes typical skincare along with topical and systemic treatment. Aside from preventive measures for AD and topical therapy, there are standard conventional systemic therapies (cyclosporine, methotrexate, azathioprine) and ongoing developments of other systemic medications, including biological agents and JAK inhibitors [9] (Table 1).

### 5.1. Preventive Measures and Emollients

Skincare is the basis of treatment for AD patients, and it should be carried out continuously, i.e., during both the phases of deterioration and the resting stages of the disease. Thus, prevention of exacerbations includes various methods and interventions (e.g., allergen avoidance, protective cream use, wearing soft clothing, using a humidifier, and others), where education and other measures play crucial roles. The basic treatment for AD includes moisturizers/emollients and baths. Due to sufferers’ deficit in cutaneous corneal lipids (especially ceramide) and “natural moisturizing factor” (a mix of amino acids resulting from filaggrin breakdown), emollients/moisturizers which include those ingredients are helpful for AD patients [40]. Skin hydration is crucial for AD patients, so emollients should be used frequently (≥2 times daily), especially after contact with water (e.g., immediately after bathing or handwashing). Thick creams or ointments (e.g., petroleum jelly) without water are preferable because they are more effective. For severe AD manifestations, wet dressings (wet wraps) may be used to help soothe the skin, decrease itch, and stop the itch–scratch cycle [40].

### 5.2. Topical Steroids, Topical Immunomodulators, and Other Topical Agents

Topical corticosteroids remain the gold standard of AD treatment as the quickest and most effective at calming and controlling skin inflammation. The strength of the corticosteroids used is based on the severity of the disease and patient age. The frequency of their use varies but is usually relatively short to avoid the *rebound phenomenon*. Other adverse effects of prolonged use include skin atrophy (thinning), telangiectasias, folliculitis, striae, purpura, and contact dermatitis. In addition, long-term topical corticosteroid use (especially high- or super-high-potency preparations) on large areas can lead to adrenal suppression [40]. Thus, “corticophobia” is common among patients. Still, a proactive therapy regimen primarily using topical corticosteroids is important. It is very useful in the maintenance and prevention of relapses.

When topical corticosteroids cannot be applied over a longer time due to thin skin (e.g., eyelids) or the occlusive effect in skin folds (skin folds including axillae and groins, anogenital area), the topical immunomodulators tacrolimus and pimecrolimus are recommended. They exert an anti-inflammatory effect by inhibiting calcineurin in skin cells and, unlike corticosteroids, do not cause side effects (e.g., skin atrophy). They are indicated for ameliorative phases and as proactive therapy.

Therapy sometimes depends on the localization of AD lesions. For lesions on the trunk and limbs, thicker emollients and moderate-potency topical corticosteroid fatty preparations are useful, with potent topical steroids for flare-ups. Sometimes pulse therapy is possible, when the potent preparation is prophylactically used two consecutive days per week. For facial lesions, light emollients are recommended, while topical corticosteroids such as *hydrocortisone* cream are effective on active lesions. For maintenance therapy, pimecrolimus is used, and for severe facial manifestations, moderate-potency topical corticosteroids are approved for short-term use. For lesions on the hands and feet and for very thick stratum corneum (palmoplantar sites), ultrapotent topical corticosteroids may be given. For axillary and groin lesions, hydrocortisone or pimecrolimus is usually adequate (flexures generally do not require emollients, and potent topical steroids may only sometimes be used for a short period of time).

AD patients are at increased risk for associated cutaneous infections, predominantly bacterial, but also viral and fungal infections [40]. *Staphylococcus aureus* (*S. aureus*) is a frequent skin colonizer in AD patients (as recorded in 70% of AD lesions and 39% of nonlesional skin of AD patients), although this bacterium’s role in driving AD severity is still unclear. In addition, there is a large body of evidence supporting a relationship between heavy *S. aureus* colonization and eczema severity, as the pooled colonization rate is higher in patients with severe AD than in those with mild AD (83%:43%) [40]. Concerning therapy, topical mupirocin cream is useful for patients with localized bacterial infections; however, prolonged topical antibiotic use can lead to therapy-resistant strains and should be avoided. In more extensive infections, a two-week treatment with oral antibiotics such as cephalosporins or penicillinase-resistant penicillins is recommended. Since sodium hypochlorite has anti-staphylococcal properties (6% solution) and also has an effect on methicillin-resistant *Staphylococcus aureus* (MRSA), diluted bleach baths are an adjunct to topical therapy between episodes of infection [40]. For viral infections, lesions with herpes simplex are treated with oral antiviral therapy, and intravenous antiviral therapy can sometimes be used in severe cases. Concerning fungal infections, dermatophyte infections are the most frequent in AD patients and can be treated with standard antifungal therapy (in patients with head/neck manifestations of AD, commensal *Malassezia furfur* yeast may trigger flare-ups).

Other topical preparations mentioned in the literature are crisaborole, a topical phosphodiesterase 4 (PDE4) inhibitor (a topical selective JAK inhibitor ruxolitinib 1.5% cream, approved by the FDA), as well as topical doxepin, a tricyclic antidepressant with potent H1- and H2-blocking properties.

In the future, it is expected that new topical drugs will become available, but currently, no drug has proven superior in effect to topical corticosteroids.

### 5.3. Phototherapy and Conventional Systemic Therapy

Phototherapy for AD is predominantly narrow-spectrum UVB phototherapy; thus, the UVB rays penetrate the epidermis and up to the upper/top dermal layer, where they are absorbed by the cells. It has the effect of altering DNA expression and the release of anti-inflammatory cytokines, which has an immunosuppressive effect on the skin’s immune system. Phototherapy usually requires 3–5 patient visits per week to facilitate where phototherapy is used until a certain number of irradiations have been completed (usually around 20), which requires time and discipline.

Systemic AD therapy includes systemic corticosteroids, phototherapy, conventional systemic treatment (cyclosporine, methotrexate, azathioprine), and recently, new, specific drugs, interleukin inhibitors and JAK inhibitors [41,42,43,44,45,46,47,48,49,50,51,52,53,54,55,56,57,58]. Systemic corticosteroids are only used for a short time in the acute stages, and it is necessary to avoid using them over a longer period of time.
life-13-01419-t001_Table 1Table 1Prominent systemic drugs for atopic dermatitis and their characteristics.Systemic Drugs for Atopic DermatitisMechanisms (in Atopic Dermatitis)Adverse ReactionsSystemic corticosteroidsGlucocorticoids act by binding to and activating intracellular glucocorticoid receptors. When activated, glucocorticoid receptors bind to regions of DNA responsible for the promotion and activation of transcription factors, which then results in the inactivation of genes. Methylprednisolone is a powerful medication with anti-inflammatory properties and the ability to inhibit the immune system.Congestive heart failure in susceptible patients, hypertension, fluid retention, muscle weakness, loss of muscle mass, steroid myopathy, osteoporosis, tendon rupture, vertebral compression fractures, aseptic necrosis of femoral and humeral heads, pathologic fracture of long bones, peptic ulcer with possible perforation and hemorrhage, pancreatitis, abdominal distention, ulcerative esophagitis, impaired wound healing, petechiae and ecchymoses, may suppress reactions to skin tests, thin and fragile skin, facial erythema, increased sweating, increased intracranial pressure with papilledema (pseudo-tumor cerebri), convulsions, vertigo, headache. Cushingoid state, growth suppression in children, secondary adrenocortical and pituitary unresponsiveness, menstrual irregularities, decreased carbohydrate tolerance, latent diabetes mellitus, increased requirements of insulin or oral hypoglycemic agents in diabetics. Posterior subcapsular cataracts, increased intraocular pressure, glaucoma, exophthalmos. Laboratory values: sodium retention, potassium loss, hypokalemic alkalosis, increases in ALT, AST, and ALP, negative nitrogen balance.CyclosporineCyclosporine has potent immunosuppressive properties. Cyclosporine inhibits cell-mediated reactions and T-cell-dependent antibody production. Cyclosporine inhibits the production and release of lymphokines including I- 2 at the cellular level. Anorexia, tremor, headache, convulsions, paraesthesia, hypertension, flushing, nausea, vomiting, abdominal discomfort/pain, diarrhea, gingival hyperplasia, peptic ulcer, hirsutism, acne, hypertrichosis, myalgia, muscle cramps, pyrexia, fatigue.Laboratory values: leucopenia, hyperlipidemia, hyperglycemia, hyperuricemia, hyperkalemia, hypomagnesemia, hepatic function abnormal, renal dysfunction.MethotrexateMethotrexate has the capacity to decrease inflammation and suppress an overactive immune system. Methotrexate is a folic acid antagonist, with antimetabolite properties. It inhibits DNA synthesis by the competitive inhibition of the enzyme dihydrofolate reductase.Headache, tiredness, drowsiness, pneumonia, interstitial alveolitis/pneumonitis often associated with eosinophilia, stomatitis, dyspepsia, nausea, loss of appetite, abdominal pain, oral ulcers, diarrhea, exanthema, erythema, pruritus.Laboratory values: leukopenia, anemia, thrombopenia, abnormal liver function tests (increased ALT, AST, ALP, and bilirubin).AzathioprineAzathioprine is an imidazole derivative of 6-mercaptopurine (6-MP). Azathioprine is broken down into 6-MP and a methylnitroimidazole. The 6-MP is converted in the cell into several purine thioanalogs. Viral, fungal, and bacterial infections in transplant patients receiving azathioprine in combination with other immunosuppressants, bone marrow depression, nausea.Laboratory values: leukopenia, thrombocytopenia.Mycophenolic acidMycophenolate mofetil has immunosuppressive properties. It is converted into mycophenolic acid which blocks an enzyme called “inosine monophosphate dehydrogenase”. The enzyme inosine monophosphate dehydrogenase is important for the formation of DNA in cells, especially in lymphocytes. Consequently, it reduces the rate of lymphocyte multiplication.Bacterial infections, fungal infections, viral infections, benign neoplasm of skin, neoplasm, skin cancer, ecchymosis, pseudolymphoma, gout, weight loss, confusion, depression, insomnia, agitation, anxiety, cognitive impairment, dizziness, headache, hypertonia, paresthesia, somnolence, tremor, convulsion, dysgeusia, tachycardia, hypertension, hypotension, lymphocele, venous thrombosis, vasodilatation, cough, dyspnea, pleural effusion, abdominal distension, abdominal pain, colitis, constipation, decreased appetite, diarrhea, dyspepsia, esophagitis, eructation, flatulence, gastritis, gastrointestinal hemorrhage, gastrointestinal ulcer, gingival hyperplasia, ileus, mouth ulceration, nausea, pancreatitis, stomatitis, vomiting, hypersensitivity, hepatitis, jaundice, acne, alopecia, common rash, skin hypertrophy, arthralgia, muscular weakness, asthenia, chills, edema, hernia, malaise, pain, pyrexia, hematuria, renal impairment. Laboratory values: anemia, leukocytosis, leukopenia, pancytopenia, thrombocytopenia, acidosis, hypercholesterolemia, hyperglycemia, hyperkalemia, hyperlipidemia, hypocalcemia, hypokalemia, hypomagnesemia, hypophosphatemia, hyperuricemia, hyperbilirubinemia, increased alkaline phosphatase, lactate dehydrogenase, hepatic enzyme, creatinine, blood urea.DupilumabDupilumab is a monoclonal antibody. It is designed to block receptors for IL-4 and IL-13. It relieves symptoms by blocking receptors and preventing IL-4 and IL-13 from working. Patients suffering from AD have high levels of IL-4 and IL-13, which can cause inflammation of the skin leading to symptoms.Conjunctivitis, oral herpes, conjunctivitis allergic, arthralgia, injection site reactions (includes erythema, edema, pruritus, pain, and swelling).Laboratory values: eosinophilia.AbrocitnibAbrocitinib blocks the action of Janus kinase enzymes which have an important role in inflammation that occurs in AD. Abrocitinib reduces inflammation and itching of the skin.Herpes simplex, herpes zoster, headache, dizziness, nausea, vomiting, upper abdominal pain, acne.Laboratory values: increased CPK.BaricitinibBaricitinib is an immunosuppressant medication and works by disabling the Janus kinase enzymes which have an important role in the processes of inflammation and damage that occur in patients with AD. Baricitinib reduces skin inflammation and symptoms of AD.Upper respiratory tract infections, herpes zoster, herpes simplex, gastroenteritis, urinary tract infections, pneumonia, folliculitis, headache, nausea, abdominal pain, rash, acne. Laboratory values: thrombocytosis, hypercholesterolemia, increased ALT, CPK.UpadacinibUpadacitinib has immunosuppressant properties and acts by blocking the Janus kinase enzymes which are involved in the processes that lead to inflammation, and blocking them controls symptoms.Upper respiratory tract infections, bronchitis, herpes zoster, herpes simplex, folliculitis, influenza, urinary tract infection, cough, abdominal pain, nausea, acne, urticaria, rash, fatigue, pyrexia, weight increased, headache.Laboratory values: anemia, neutropenia, lymphopenia, hypercholesterolemia, hyperlipidemia, increased CPK, ALT, AST.Abbreviations: CPK—creatine phosphokinase; ALP—alkaline phosphatase; ALT—alanine transaminase; AST—aspartate transaminase; 6-MP—6-mercaptopurine.


Conventional systemic medications for AD include immunosuppressive and immunomodulatory drugs that do not act specifically on AD but suppress the patient’s immune system and calm inflammation. Cyclosporine has been approved for treatment, while methotrexate, azathioprine, and mycophenolate mofetil are used off-label. Cyclosporine is an immunosuppressant that blocks numerous immune cells and inhibits the activation of T cells and NK cells and antigen presentation by APCs through a calcineurin-dependent pathway, as well as the production of IL-2 and GM-CSF. It is nephrotoxic and may cause hypertension, changes in differential blood counts, gingival hyperplasia, hypertrichosis, and headaches. Methotrexate is an immunomodulatory drug that comes in oral and subcutaneous forms and is taken with folic acid. It is an antifolate metabolite that blocks the synthesis of DNA, RNA, and purines, inhibiting T cells. The main side effects of methotrexate include hepatotoxicity and a reduction in the number of leukocytes and platelets in the blood. Azathioprine is a purine analog that inhibits DNA production in T and B cells. The drug is hepatotoxic and can cause leukopenia and gastrointestinal symptoms. Therefore, when using all three drugs, intensive monitoring of laboratory parameters and other clinical signs and symptoms is required.

### 5.4. Biologic Therapies and JAK Inhibitors

Recently, several specific drugs have been approved, or are about to be approved (country-specific), for treating moderate to severe clinical AD. These therapy options primarily include biological drugs that inhibit IL-4, IL-13, and IL-33 and JAK inhibitors (baricitinib, abrocitinib, upadacitinib, etc.) (Figure 2).

Among biological drugs, a breakthrough came when dupilumab, which is directed against IL-4 and IL-13 receptors, was designed and introduced [59]. Dupilumab is designed to block receptors for IL-4 and IL-13, and it is used as a modifying drug for patients suffering from moderate to severe AD for whom topical therapies are insufficient and other systemic treatments are not recommended. In March 2017, dupilumab was approved in the U.S. as first-line therapy for moderate and severe AD in adults, after which it was also approved for children and adolescents (12–18 years). In November 2020, it was also approved for severe AD in younger children (6 to 11 years). The recommended dose for adults and adolescents (>60 kg) is 600 mg subcutaneously, followed by 300 mg every second week. These patients should also use daily emollients and may use topical anti-inflammatory drugs as necessary [9]. The most common side effects include conjunctivitis (usually temporary), ocular complications, and skin reactions at the site of application [59].

There are also three Janus kinase (JAK) inhibitors approved for the treatment of moderate to severe AD: baricitinib, upadacitinib, and abrocitinib. They specifically and simultaneously inhibit inflammatory pathways, thus improving the patient’s condition. Janus kinase inhibitors are intracellular enzymes that transduce signals (which result from interactions between cytokines or growth factors and cell receptors) to influence hematopoiesis and immune cell functioning. Janus kinase inhibitors phosphorylate and activate Signal Transducers and Activators of Transcription (STATs) that modulate intracellular activity, including genetic expression. When taking into consideration contraindications, necessary laboratory controls, and side effects, JAK inhibitors represent a highly effective and safe treatment option for patients with severe AD [60]. According to clinical trials and the first real-life data results, JAK inhibitors offer rapid and effective itch reduction and condition improvement, leading to a better quality of life. The safety profile is favorable, although data on long-term safety, especially for the treatment of AD patients, are still lacking [60]. Other JAK inhibitors are tofacitinib, ruxolitinib, and deucravacitinib, but limited data on their use and effectiveness are available.

In October 2020, the European Medicines Agency (EMA) approved baricitinib for moderate and severe AD in adults. Baricitinib was the first approved oral selective inhibitor of JAK-1 and JAK-2. Baricitinib is a selective and reversible inhibitor of JAK1 and JAK2, which inhibits inflammatory processes via intracellular actions [61]. The recommended dose of baricitinib is 4 mg peroral once daily (or less as needed). Adverse reactions are respiratory infections, hypercholesterolemia, herpes infections, uroinfections, gastroenteritis, and thromboembolism, among others [61]. Previous study results (BREEZE-AD1, BREEZE-AD2, BREEZE-AD7) have shown significant improvements in AD patients’ clinical pictures [62]. Upadacitinib is also an oral selective JAK1 inhibitor, and the EMA approved it in August 2021 for the treatment of moderate to severe AD, for children older than 12, adolescents, and adults [63]. The generally recommended dose for AD is 30 mg daily, which quickly and significantly improves the patient’s condition, even within the first week [63]. Upadacitinib has been shown to be safe and efficacious in several clinical trials. Common adverse events are respiratory infections, herpes zoster infections, blood test abnormalities (increase in blood creatine phosphokinase (CPK))*,* papulopustular acne, and others [64]. In addition, abrocitinib inhibits JAK1, which modulates multiple cytokines involved in AD pathogenesis and manifestations (IL-4, IL-13, IL-31, IL-22, and stromal timus lymphopoietin). It is used for moderate and severe AD in those older than 12. Abrocitinib was approved in December 2021 in Europe as a peroral drug (100 mg and 200 mg daily). Common adverse reactions are nausea and respiratory infections.

In addition, the biological drug tralokinumab has recently been approved for AD. Tralokinumab is the first fully human monoclonal antibody that binds specifically to IL-13 and which has been shown to be safe and effective in clinical trials. In combination with topical corticosteroids, tralokinumab provides quick and persistent AD improvement, as exhibited in patients’ clinical pictures. The most frequent adverse events are AD flare-ups, skin reactions at the injection site, respiratory infections, and conjunctivitis [65]. Other biological agents mentioned in the literature are nemolizumab and mepolizumab, but only limited study results are available.

Despite these therapy options, the high cost of newer drugs is often a barrier to treatment access, which is why some countries are not able to purchase biological agents such as dupilumab. Since healthcare costs and spending are generally high and increase over time, data on affordability and access to new treatments, as well as the economic consequences of treatment costs, needs to be systematically documented and analyzed. In addition, AD is clinical condition that significantly affects patients’ quality of life and requires continuous care [66]. Various measures (indices) (EASI, DLQI, NRS) can be used to determine disease severity and the positive influence post-treatment improvement has on quality of life. These measures have shown a positive correlation between annual therapy results and clinical variables [66]. So, although these therapies are expensive, their efficacy can make them well worth the cost.

On the other hand, the use of traditional biologics (rituximab, omalizumab, or ustekinumab) as a treatment for AD cannot be recommended. In certain cases where standard therapy is ineffective, mepolizumab may be considered as a treatment option.
Figure 2Roles/activities of biologic therapies and JAK inhibitors in the pathogenetic network in atopic dermatitis—modified based on the article by Eichenfield et al. [67]. Adapted with permission from Ref. [67]. Copyright 2023, copyright Eichenfield et al.
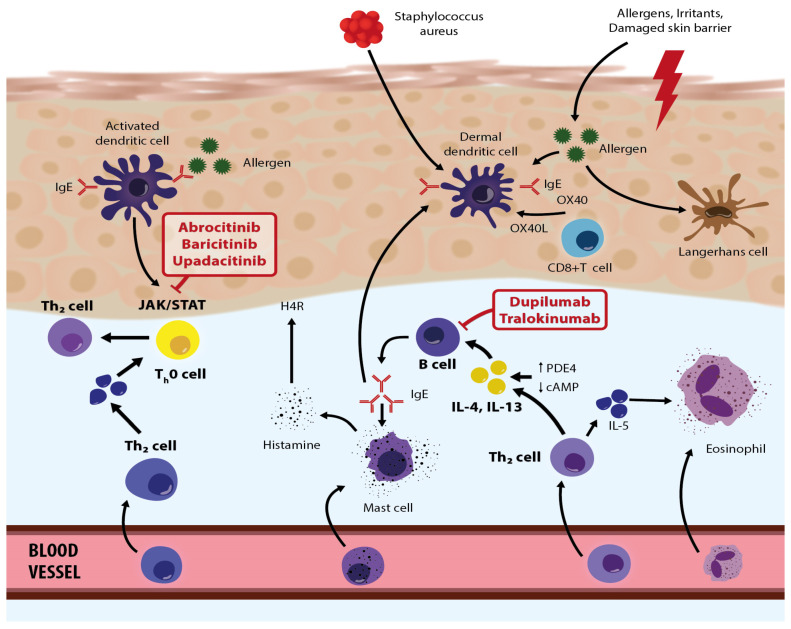



## 6. The Impact of AD on the Psychological Functioning and Quality of Life of Patients

Like other chronic recurrent diseases, AD affects the physical, psychological, psychosocial, and occupational aspects of life [68]. In addition to skin lesions, itching is a major problem for AD patients. Its intensity is primarily associated with AD severity, and it commonly disrupts the sleep cycle and causes problems with sleeping (e.g., changes in melatonin values were recorded) [69,70]. Emotional factors can trigger or exacerbate the itching and scratching, significantly affecting patients’ quality of life [68]. Patients often report a link between emotional stress, itching, and scratching. Thus, itching itself is a source of permanent psychological distress for the patient, and of all of the clinical AD symptoms, itching is most strongly associated with anxiety [71,72]. Thus, a higher incidence of anxiety and depression is observed in AD patients [73,74,75], and almost all patients with moderate to severe AD have symptoms of anxiety and depression, which should be considered when assessing an AD patient’s condition [74]. In studies, AD patients show higher levels of anxiety, anger, and worry and have a lower quality of life than healthy individuals and higher levels of anxiety than patients with psoriasis [75,76,77,78,79]. Several studies have also shown an increased risk of suicidal thoughts in adult AD patients [75,77].

Thus, one of the commonly reported triggers and disease-related factors for AD patients is psychological stress. Suffering from AD is a proven stressor, primarily due to frequent and unexpected AD exacerbations, and patients commonly mention psychological stress as a trigger for their disease. It is useful, then, to assess patients’ perception of stress, the Perceived Stress Scale (PSS) being the common tool for this [69,70,71,72]. According to recent research results concerning AD patients, the level of perceived stress correlates with the impact of the disease on emotional states and personality features (anxiety, depression) [80]. According to research data, stress participates in/influences the disruption of epidermal barrier permeability homeostasis and may alter neurogenic and humoral immunity via the central and peripheral HPA axis and autonomic nervous system. Elevated glucocorticoid and catecholamine levels inhibit the secretion of IL-12 and promote a shift towards a Th2 inflammatory response, which is crucial in AD pathogenesis and manifestations. Psychological stress and stress hormones increase the synthesis of the neurotransmitter serotonin, which has a pro-inflammatory effect locally on the skin (edema, vasodilatation, and itching), and its receptors are located on keratinocytes, melanocytes, and dermal fibroblasts [72]. In addition, cutaneous sensory nerve fibers release substance P, CGRP, and nerve growth factor (NGF), which lead to pain and itching. Mast cells are the central cells of humoral immunity (they are activated by various mediators of stress hormones), expressing different neuropeptide/neurohormone receptor isoforms, including CRF [81]. In addition, mast cells synthesize and secrete more than 50 biologically active molecules that mediate neurogenic inflammation, including substance P, serotonin, TNF-α, NGF, tryptases and chymases, and histamine. Such findings open up the possibility for new psychological therapeutic strategies.

The results of several studies support patients’ reports that AD strongly influences their quality of life, especially psychological well-being and social functioning. The most commonly used quality-of-life tests in dermatology are the Dermatology Life Quality Index (DLQI) and the Skindex-16 [78]. These tests are able to measure AD patients’ quality-of-life impairment [68,69,82,83]. Their quality of life is associated with lower overall health, and life satisfaction ratings weaken the dermatological quality of life and quality of life related to mental health. Generally, a greater severity of the clinical picture correlates with a decrease in quality of life [68,69]. Thus, moderate and severe forms of AD are associated with a significantly lower quality of life, as seen in patients with other chronic diseases (such as diabetes, heart disease, and hypertension) [83]. When determining the impact of AD on quality of life, psychological functioning, and health, numerous psychometric tests are used to assess/measure the patient’s perception of physical, emotional, social, environmental, and other factors that may affect, or be a consequence of, the disease. The impact of AD on quality of life is unexpectedly strong, and patients often have various accompanying problems (e.g., sadness, stress during coping with their disease, functional impairments). According to recent research data, AD patient quality of life depends, aside from AD severity, on personality characteristics (anxiety, obsession, depression, somatization), and clinicians should take this into account [84]. Notably, there is often no correlation between a patient’s belief about their disease condition and the actual condition/severity of the disease, so the patient’s perception is strongly influenced by the importance they attribute to their condition rather than the objective disease severity [69].

## 7. Multidisciplinary Approach to Patients with Atopic Dermatitis

Overall, the current approach to patient management and the treatment of AD is costly and complex for doctors, patients and their families, and society [68]. This is partly due to the fact that many AD patients have various concomitant disease-related issues that sometimes overlap, meaning specialists from various occupations are often involved in coordinated therapy regimens, such as psychologists, ear–nose–throat specialists, pulmonologists, allergologists, immunologists, nutritionists, pediatricians, gastroenterologists, and psychiatrists (when necessary) (Figure 3) (Table 2) [9,85,86,87,88,89,90,91,92,93]. Therapy usually involves medical evaluations and management, patient education and nursing care, psychological support, and nutritional consultations, which are organized through specific programs and structured educational groups [85,86,87]. For example, since “corticophobia” is common among patients with AD and parents of children with AD, educational programs on the effects and side effects of topical corticosteroids can help patients overcome their hesitation in seeking treatment, thus improving their condition and quality of life.

The majority of results from studies on a multidisciplinary approach to AD have observed improvements in patients’ clinical conditions and their concomitant problems and manifestations. However, not all results were consistent, as some studies did not find clear benefits to a multidisciplinary approach [86]. Concerning quality of life, findings regarding the impact of multidisciplinary interventions were mixed, likely due to differences in methodology across studies. Some studies assessing interventions for children with AD reported improved quality of life from the parents’ perspective, which reflects the positive impact multidisciplinary interventions can have on family quality of life [86]. In addition, it is useful to consider that certain patients may particularly benefit from a multidisciplinary approach. For example, educational training programs generally serve patients with moderate/severe AD; however, other AD patients also benefit from such programs—those who fail to respond to conventional medical treatments, who are concerned about side effects from treatment, who report a significant impact on quality of life, who experience recurrent skin infections, who avoid many foods due to allergy concerns, etc. [86]. Multidisciplinary care is also easily adjusted to optimize treatment based on patient age and dermatitis severity.

A multidisciplinary approach provides better coping strategies and improves control over the disease, patient adherence to therapy, and quality of life. In other words, it positively impacts the patient’s condition and outcome (disease severity, itch), and it also has a positive influence on family quality of life while at the same time making more efficient use of dermatology healthcare resources, reducing the economic burden on both patients and society. Current literature data clearly show that a multidisciplinary approach is indeed possible and necessary and that it provides the best outcome in the end for those living with AD [85,86,87]. Therefore, the multidisciplinary approach should be included in the universal guidelines for AD management, as has already been done in some countries. 

## 8. Conclusions

Patients with AD often face a range of disorders and difficulties that can vary considerably in nature and severity. The results of many studies confirm AD’s strong influence on patient quality of life, including daily activities, psychological well-being, and social functioning. Thus, a multidisciplinary approach is the most effective and important for AD patients, where, aside from dermatologists, different professions are involved, such as psychologists, ear–nose–throat specialists, pulmonologists, nutritionists, and pediatricians. Since AD is a chronic disease, many AD patients are affected by various psychological factors and comorbidities, and they should be assessed and managed through a multidisciplinary approach. This is the best way to simultaneously manage various aspects of the disease.

## Figures and Tables

**Figure 1 life-13-01419-f001:**
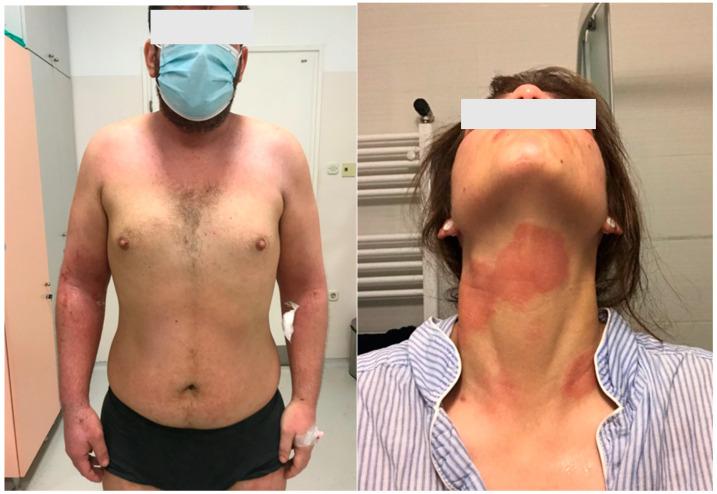
Clinical pictures of the patients with atopic dermatitis.

**Figure 3 life-13-01419-f003:**
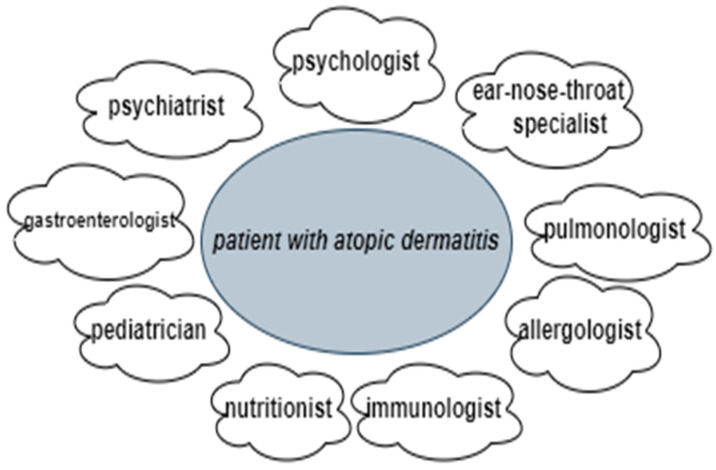
A multidisciplinary approach to a patient with atopic dermatitis.

**Table 2 life-13-01419-t002:** Roles of various specialties and their activities in a multidisciplinary approach to AD patients.

Specialists	Roles and Activities in a Multidisciplinary Approach to AD Patients
Psychologists (Kauppi et al., 2019) [88]	Psychological manifestations of pruritus and scratching, such as sleep disruptions and decreased quality of life, are common and sometimes severe. Sleep disturbance is one of the most burdensome symptoms reported by AD patients (recorded in 33–87.1% of adult patients). A higher incidence of depression and anxiety among patients with AD was confirmed. →Approach (psychological) includes giving clinical advice, clinical examinations, and management (e.g., biofeedback, cognitive behavioral therapy, stress management).
Ear–nose–throat specialists (Lee SW et al., 2023) [89]	Associated/concomitant allergic rhinitis or allergic sinuitis are relatively common in AD patients. AD is found to be associated with an increased risk of subsequent *gastroesophageal reflux disease* (GERD); appropriate management should be considered in adults with AD to prevent GERD. →Approach includes clinical examinations and management by ear–nose–throat specialists.
Pulmonologists (Kansen et al., 2020) [90]	Due to the association between atopic diseases (asthma, allergic rhinitis, AD, food allergy) and recurrent respiratory tract infections, especially in children, an examination of the respiratory tract is often needed. Children with atopic diseases may be more susceptible to respiratory infections, probably caused by airway inflammation and an innately impaired immune system (sometimes infections may actually stimulate a patient’s protective immunity, thereby reducing the risk of atopic diseases). →Approach includes clinical examinations and management by a pulmonologist.
Allergologists (Wollenberg et al., 2018) [9]	AD flare-ups may be elicited by various allergens or by irritants (e.g., wool), chemicals (acids, bleaches, solvents, water) or other factors (e.g., microbes). Air pollutant (tobacco smoke or volatile organic compounds), as well as traffic exhaust, exposure is potentially connected to AD exacerbations in young children. The abovementioned compounds disrupt skin barrier functions and may aggravate AD. →Approach includes clinical examinations and management by an allergologist.
Immunologists and infectologists (Wollenberg et al., 2018) [9]	AD patients are at increased risk for cutaneous infections, predominantly bacterial, but also viral and fungal infections. *Staphylococcus aureus*, which heavily colonizes AD lesions, produces extracellular proteases, which cause skin barrier breakdown and thus facilitate allergen uptake and specific sensitization; excoriations and secondary infections occur due to scratching and disruption of the skin barrier. Some issues belong to the field of immunology, e.g., AD worsening connected to vaccination or animals in the house. If dogs live in a household, however, though increased exposure to bacteria exists, this may have a protective effect via primary prevention and immune regulation, or it can include a risk for bacterial superinfection (when skin lesions are in contact with *dogs*). Considering vaccination, all children diagnosed with AD should be vaccinated according to the local or national vaccination plan. →Approach includes giving clinical advice, examinations, and management. Avoidance of irritating fabrics and fibers and relevant contact allergens is essential. This is especially true if the contact allergy is to ingredients of emollients and has been confirmed by patch tests.
Nutritionists (Khan A et al., 2022). [91]	Environmental triggers such as food/dietary exposures contribute to AD manifestations. Poor diet is believed to be associated with an increased risk of AD exacerbations. Diet in pregnancy and early infancy is consequential to AD manifestations. Though not confirmed, some factors may positively influence manifestations/prevention (maternal dietary restrictions during pregnancy/lactation, hydrolyzed/partially hydrolyzed formulas, delaying the introduction of solid foods, and omega-3/omega-6 fatty acids supplementation, breastfeeding for at least 3 to 4 months after birth, a diet high in fruits and vegetables, and the use of prebiotics). In infancy and adulthood, there are also some food allergens connected to AD exacerbations, such as soy, milk, peanuts, corn, carrots, rye, wheat, egg white, cod, and chicken. →Approach includes clinical examinations and management by a nutritionist.
Pediatricians (Tracy A et al., 2020) [92]	AD is especially common in infants and children and mostly impacts quality of life and sleep, theirs and their parents’/guardians’. Bathing and moisturizing regimens differ tremendously from those of healthy children and require much more attention, both from the child and their parents. AD also affects school performance and mood, which indirectly affects children’s success in school and social life. →Approach includes clinical examinations and management by a pediatrician.
Gastroenterologists (Islamoǧlu, Z. K. et al., 2019) [93]	The most common AD-associated gastrointestinal disorders are abdominal pain vomiting, diarrhea, abdominal distention, and constipation. There is also a link to functional gastrointestinal disorders such as irritable bowel syndrome (IBS), which is significantly higher in AD patients. The incidence of gastrointestinal disorders does not depend on AD extent and severity. Gastrointestinal disorders manifest independently of the type of allergic reactions (inducing AD). It is well known that there is a skin–gut–lung connection via immunologic dysregulation and microbiome alterations in atopic diseases. Patients with AD should be examined for IBS. →Approach includes clinical examinations and management by a gastroenterologist.
Psychiatrists (Kauppi et al., 2019) [88]	AD patients are prone to a higher incidence of depression and anxiety, which are associated with an autonomic dysbalance with relatively decreased vagal activity. At least one psychiatric diagnosis was found in 17.2% of AD patients and in 13.1% of controls. Psychotherapy provides beneficial effects related to pruritus and scratching, and its positive effect on patient condition has been observed in clinical practice. Psychiatric morbidity is significant in AD patients; therefore, assessing patients’ mental health should be considered a part of standard care. Psychotropics can be useful, but they should be used with caution in infants. →Approach includes clinical examinations and management by a psychiatrist.

Abbreviations: AD—atopic dermatitis; GERD—*gastroesophageal reflux disease*; IBS—irritable bowel syndrome.

## Data Availability

The data that support the findings of this study are openly available in PubMed or available in other sources.

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
