# Peer review of "Atopic Dermatitis: Disease Features, Therapeutic Options, and a Multidisciplinary Approach"

_life, 2023, doi:10.3390/life13061419_

Round 1

Reviewer 1 Report

The manuscript is very informative and elaborated on the aimed topics in the title. I can recommend just addition of the wrod" dermatologist" in the central circle in the fig. 2 

Author Response

Thank You for Your valuable time, effort and useful contribution You have put into assesing our previous version of our manuscript. We really appreciate the input You have given because it has definitely improved our manuscript.

We have given careful consideration to each comment. 

Reviewer 2 Report

The manuscript entitled ‘Atopic Dermatitis: Disease Features And a Multidisciplinary Approach’ is a review paper which is focused on the latest findings in pathogenesis, clinical manifestations and diagnostic criteria of AD. In addition there is a chapter concerning treatment methods and the impact of quality of life of the patients with AD.

The manuscript is interesting, however it would benefit from a revision.

1.     I don’t exactly see the main goal of this manusctript. Authors have to decide what is the main topic of this paper and describe it in details in the body of the manuscript according to the title.

There are a lot of reviews and books about AD published in recent years, so I don’t see any novelty in this exact manuscript. Maybe the authors should focus more on the multidisciplinary approach with a deep review of literature in this exact field. In actual manuscript there is not even a separate chapter/section on this topic – it is only mentioned in the section 6 which is devoted to the psychological functioning and quality of life of patients.

2.     Figure 1 - there is no information about the source of photos included in the manuscript. In addition there is inadequate protection of anonymity of the patients. I would also recommend using this figure in the section 4 – Clinical manifestations of AD, not introduction.

3.     Section 4 – Clinical manifestations of atopic dermatitis and diagnostic criteria.

·       There is not enough information about the clinical features of AD. The authors should describe in details what can be seen on patient’s skin (xerosis, lichenification), what are the main and additional features of AD. In my opinion this section is not detailed enough.

·       Line 250 – please provide the correct citation

4.     Section 5 – Available treatment possibilities for atopic dermatitis.

First of all, I would recommend division this chapter into sections (e.g. topical steroids, immunomodulators, biologic therapies, JAK inhibitors etc.). It will be easier for readers to follow.

·       There is not enough information about the basic treatment which is: moisturisers/emollients/baths. Neither we can find any information on the prevention of exacerbations of the disease (such as nonmedical interventions: soft clothing, humidifier etc.), education.

·       The authors did not mention antibiotics for clinical infections caused by S. aureus or flares of the disease (although this is the special issue: Microbiome, Microorganisms and Skin in the Section of Microbiology) or viral superinfections - maybe this is an interesting topic that should be described in this manuscript as a leading one with consideration of multidisciplinary approach of patients with AD.

·       The authors write that we can find information about ongoing developments of other oral medications (line 357) in Table 1, whereas table 1 includes partly the same medications which were described above and are widely known (e.g. cyclosporine, MTX).

5.     There are some small literal errors, e.g. dots in the middle of the sentence – line 350 and others.

6.     I would also recommend adding more schemes, figures which will help readers systematize the data included in the text (e.g. pathophysiology).

 I suggest an English revision – a few sentences are not clear, e.g. line 420: Using those tests many studies show have shown impairment in AD patients’ quality of life or line 444: Due to visible their skin lesions (…). In addition, articles (a, an, the) should be corrected.

Author Response

REVIEWER 2

The manuscript entitled ‘Atopic Dermatitis: Disease Features And a Multidisciplinary Approach’ is a review paper which is focused on the latest findings in pathogenesis, clinical manifestations and diagnostic criteria of AD. In addition there is a chapter concerning treatment methods and the impact of quality of life of the patients with AD. The manuscript is interesting, however it would benefit from a revision.

  1. I don’t exactly see the main goal of this manuscript. Authors have to decide what is the main topic of this paper and describe it in details in the body of the manuscript according to the title. There are a lot of reviews and books about AD published in recent years, so I don’t see any novelty in this exact manuscript. Maybe the authors should focus more on the multidisciplinary approach with a deep review of literature in this exact field. In actual manuscript there is not even a separate chapter/section on this topic – it is only mentioned in the section 6 which is devoted to the psychological functioning and quality of life of patients.

  Thank you for your suggestions, we changed the text and added more data (with additional text) on multidisciplinary, including a new table with more data on this topic.

  1. 2. Figure 1 - there is no information about the source of photos included in the manuscript. In addition there is inadequate protection of anonymity of the patients.  I would also recommend using this figure in the section 4 – Clinical manifestations of AD, not introduction.

    According to these suggestions, we moved the figures to this another part of this manuscript. Here are the pictures of our two patients, obtained with their aprovals.

  1. Section 4 – Clinical manifestations of atopic dermatitis and diagnostic criteria.· There is not enough information about the clinical features of AD. The authors should describe in details what can be seen on patient’s skin (xerosis, lichenification), what are the main and additional features of AD. In my opinion this section is not detailed enough.

Thank you for your suggestions, we added more information and detailed data about the clinical features of AD.

  • Line 250 – please provide the correct citation

Thank you for your suggestions, we added specific references for these citations.

  1. Section 5 – Available treatment possibilities for atopic dermatitis. First of all, I would recommend division this chapter into sections (e.g. topical steroids, immunomodulators, biologic therapies, JAK inhibitors etc.). It will be easier for readers to follow.

Thank you for your suggestions, we divided this part into the specific sections.

  • There is not enough information about the basic treatment which is: moisturisers/emollients/baths. Neither we can find any information on the prevention of exacerbations of the disease (such as nonmedical interventions: soft clothing, humidifier etc.), education.

According to these suggestions, we changed the text and added more data on the basic treatment: moisturisers/emollients/baths. the prevention of exacerbations of the disease, education, etc.

  • The authors did not mention antibiotics for clinical infections caused by S. aureus or flares of the disease (although this is the special issue: Microbiome, Microorganisms and Skin in the Section of Microbiology) or viral superinfections - maybe this is an interesting topic that should be described in this manuscript as a leading one with consideration of multidisciplinary approach of patients with AD.

Thank you for your suggestions, we added more data on infections and their treatment in AD.

  • The authors write that we can find information about ongoing developments of other oral medications (line 357) in Table 1, whereas table 1 includes partly the same medications which were described above and are widely known (e.g. cyclosporine, MTX).

   Thank you for your observation, we changed/corrected the text and added specific data (more dana) on ongoing developments of other oral medications

  1. There are some small literal errors, e.g. dots in the middle of the sentence – line 350 and others.

   Thank you for your suggestions, we found some typos and corrected them.

  1.    I would also recommend adding more schemes, figures which will help readers systematize the data included in the text (e.g. pathophysiology).

   According to these suggestions, we added one additional table and a figure.

Comments on the Quality of English Language.  I suggest an English revision – a few sentences are not clear, e.g. line 420: Using those tests many studies show have shown impairment in AD patients’ quality of life or line 444: Due to visible their skin lesions (…). In addition, articles (a, an, the) should be corrected.

According to these suggestions we changed the text to be more clear, as well as corrected some mistakes.

Reviewer 3 Report

This is an interesting review on all aspects of atopic dermatitis. Although the majority of the findings are not completely new, I think this review could be useful for many readers, as it provides a comprehensive evaluation of this disease. 

In my opinion, the section on the treatment should be improved adding some references to real-world experiences on new drugs (monoclonal antibodies and JAK inhibitors), which are important to assess the effectiveness and safety of these drugs.

- Dupilumab: 10.1080/09546634.2022.2043545

- Upadacitinib: 10.1007/s13555-022-00882-z

- Abrocitinib: 10.1007/s00105-022-05004-6

- Baricitinib: 10.2340/actadv.v102.1088

Also, the Authors should add some data on tralokinumab which is now  approved for atopic dermatitis

This manuscript is written in a very good English. 

Author Response

REVIEWER 3

This is an interesting review on all aspects of atopic dermatitis. Although the majority of the findings are not completely new, I think this review could be useful for many readers, as it provides a comprehensive evaluation of this disease. In my opinion, the section on the treatment should be improved adding some references to real-world experiences on new drugs (monoclonal antibodies and JAK inhibitors), which are important to assess the effectiveness and safety of these drugs.- Dupilumab: 0.1080/09546634.2022.2043545; - Upadacitinib: 10.1007/s13555-022-00882-z; - Abrocitinib: 10.1007/s00105-022-05004-6; - Baricitinib: 10.2340/actadv.v102.1088

Thank you for your suggestions, we added more data on these medicaments, and involved these references

Also, the authors should add some data on tralokinumab which is now  approved for atopic dermatitis

    Thank you for your suggestions, we added more data on tralokinumab.

-Comments on the Quality of English Language  This manuscript is written in a very good English. 

   Thank you for your observation!

Reviewer 4 Report

Analyzing the multidisciplinary approach that could provides better coping strategies, and improves control over the disease is very interesting. This is a review that offers some interesting points of analysis, however some issues should be explored.

1.       (Pag. 7) What do you think about the proactive therapy of topical corticosteroids? and not just by immunomodulator,

2.       (Pag. 12) it’s very important analyzing the cost and the efficacy of the therapies, to increase the discussion you could add a recent paper… “despite the fact that these terapies are expensive their efficacy can make them well worth the cost” (DOI: 10.1155/2023/4592087)

3.       In order to better understand and apply multidisciplinarity, could you explain how to organize the multidisciplinary approach for the different ages of the patients and severity of dermatitis?

4.       What about the multidisciplinary approach shoul be included in the guidelines for the management of atopic dermatitis,  it is already active in some countries?

few grammar mistakes

Author Response

REVIEWER 4

Analyzing the multidisciplinary approach that could provides better coping strategies, and improves control over the disease is very interesting. This is a review that offers some interesting points of analysis, however some issues should be explored.

 1-(Pag. 7) What do you think about the proactive therapy of topical corticosteroids? and not just by immunomodulator,

     Thank you for your suggestions, we added this information and more data on this topic to the text.

2-(Pag. 12) it’s very important analyzing the cost and the efficacy of the therapies, to increase the discussion you could add a recent paper… “despite the fact that these terapies are expensive their efficacy can make them well worth the cost” (DOI: 10.1155/2023/4592087)

   Thank you for your suggestions, we included more data related to these factors, and added this reference.

3-In order to better understand and apply multidisciplinarity, could you explain how to organize the multidisciplinary approach for the different ages of the patients and severity of dermatitis?

        According to these suggestions, we included this useful information and added more data on this subject.

       4-What about the multidisciplinary approach should be included in the guidelines for the management of atopic dermatitis,  it is already active in some countries?

        According to these suggestions, we mention that this approach is active in some countries (as in our countrs).

5-Comments on the Quality of English Language: few grammar mistakes

Thank you for your suggestions, we  corrected some mistakes and typos.

Best regards,

Authors

Round 2

Reviewer 2 Report

I would like to thank the authors for responding to my earlier comments. The manuscript is well prepared now and I feel it can be accepted for publication.